# Financial Market Correlation Analysis and Stock Selection Application Based on TCN-Deep Clustering

Yuefeng Cen [1,*] , Mingxing Luo [1], Gang Cen [1], Cheng Zhao [2] and Zhigang Cheng [1]

1   School of Information and Electronic Engineering, Zhejiang University of Science and Technology,
    Hangzhou 310023, China
2   School of Economics, Zhejiang University of Technology, Hangzhou 310014, China
*   Correspondence: cyf@zust.edu.cn

**Abstract:** It is meaningful to analyze the market correlations for stock selection in the field of financial investment. Since it is difficult for existing deep clustering methods to mine the complex and nonlinear features contained in financial time series, in order to deeply mine the features of financial time series and achieve clustering, a new end-to-end deep clustering method for financial time series is proposed. It contains two modules: an autoencoder feature extraction network based on TCN (temporal convolutional neural) networks and a temporal clustering optimization algorithm with a KL (Kullback–Leibler) divergence. The features of financial time series are represented by the causal convolution and the dilated convolution of TCN networks. Then, the pre-training results based on the KL divergence are fine-tuned to make the clustering results discriminative. The experimental results show that the proposed method outperforms existing deep clustering and general clustering algorithms in the CSI 300 and S&P 500 index markets. In addition, the clustering results combined with an inference strategy can be used to select stocks that perform well or poorly, thus guiding actual stock market trades.

**Keywords:** financial time series; clustering; deep learning; temporal convolutional neural networks





## 1. Introduction

In the field of financial investment, investors want to profit from the noisy and uncertain stock market. Most studies currently focus on the prediction of the price of a single stock or its trend classification [1–3]. However, existing studies have shown that even by using prediction models with high accuracy, it is difficult to obtain objective returns in real market environments [4,5]. As a result, investors often seek to diversify their portfolios to avoid large losses from black-swan events. Current methods for stock correlation analysis mainly mine the complex linkage of stocks through association rules, factor analysis, complex networks, etc. [6]. For example, in the field of economics, Dimitrios et al. [7] used the continuous wavelet transformation (CWT) to analyze the co-movement spillover effects, which assumes that stocks in the same industry are highly correlated and that the effects between different industries may be overlooked. However, the general approach to data mining requires a degree of competence in the financial field. Clustering algorithms have excellent results for data without prior knowledge so this study focuses on the application of clustering algorithms to the correlation of financial time series.

The general purpose of clustering analysis is to classify a set of objects into groups that are similar to each other, and it is a useful tool for exploratory data analysis in different areas of science and industry [8]. In the early days, machine learning methods based on k-means, k-shape, or spectral models could extract data features and categorize them. However, it is difficult to ensure that the features extracted by such methods fit the clustering structure of the data because the clustering results depend heavily on the way the data are represented. Current research on clustering network algorithms in finance focuses on the methods of the correlation measures for the clustering models [9] and this clustering method is also called

functional clustering. Functional clustering aims to vectorize the original data in space with a suitable metric [10–12]. Sun et al. [13] proposed a novel similarity measure based on extreme point bias compensation to measure the price similarities of SSE 50 constituents. However, this type of metric is complex and cannot be applied well in other data domains.

In recent years, deep learning has received a lot of attention for its combination with clustering tasks due to its powerful feature extraction, and representation capabilities and such methods are called deep clustering [14]. Deep clustering requires that the extracted data meet the low-dimensional representation suitable for clustering while reflecting the information characteristics and structural features of the original data [15]. Deep clustering methods have made great progress in the field of computer vision [16,17]; however, for unsupervised time-series tasks, their potential has not been fully exploited.

DEC [18] and IDEC [19] are the current novel architectures for unsupervised deep learning. These models first represent the original data via a neural network vector, and through this representation, the models then infer the data clustering class distribution. For example, in DEC, a set of raw inputs is represented by a multi-layer perceptron (MLP) to obtain a hidden vector that iteratively optimizes the clustering loss with the help of a self-learning auxiliary target distribution. IDEC adds a reconstruction term to the DEC loss function to preserve the feature space properties. Li et al. [20] proposed DBC, which replaces the pre-trained network in DEC with a CNN to extract high-quality features containing pixel space information and improve the overall clustering performance. Although the above methods significantly improve the clustering performance, in the financial field, the trends of different stocks vary widely and the fluctuations of each stock are difficult to capture [21]. The adoption of suitable characterization methods has become a challenge. Temporal convolutional neural (TCN) networks [22] were proposed by Bai et al. and are currently used in many tasks that include sequential modeling such as speech recognition, machine translation, etc. They combine the features of recurrent neural networks (RNN), which can learn historically meaningful information, and convolutional neural networks (CNN), which are computationally efficient, thus enabling the processing of sequence modeling with an advanced training speed while preserving complete temporal characteristics. This paper combines TCN networks with autoencoder networks and proposes a deep temporal clustering algorithm called TCN-Deep Clustering.

The rest of this article consists of the following: Section 2 focuses on the autoencoder temporal clustering model based on temporal convolutional neural networks. Section 3 concentrates on the experimental parameter settings and analysis of the experimental results. Section 4 validates the return performance by incorporating an inference strategy. Section 5 concludes the study and provides suggestions for other domains.

## 2. Financial Time-Series Deep Clustering Network: TCN-Deep Clustering

To address problems where general clustering methods cannot guarantee that the extracted features fit the clustering structure and current deep clustering algorithms cannot significantly capture the financial time-series features, an end-to-end deep clustering algorithm for financial time series is proposed. The architecture of TCN-Deep Clustering consists of three main components: an encoder, decoder, and temporal clustering layer, as shown in Figure 1. In the first step, a feature extraction network consisting of an autoencoder based on a TCN network is pre-trained and in the second step, the temporal clustering layer is added to fine-tune the hidden feature. The algorithm's details are described as follows.

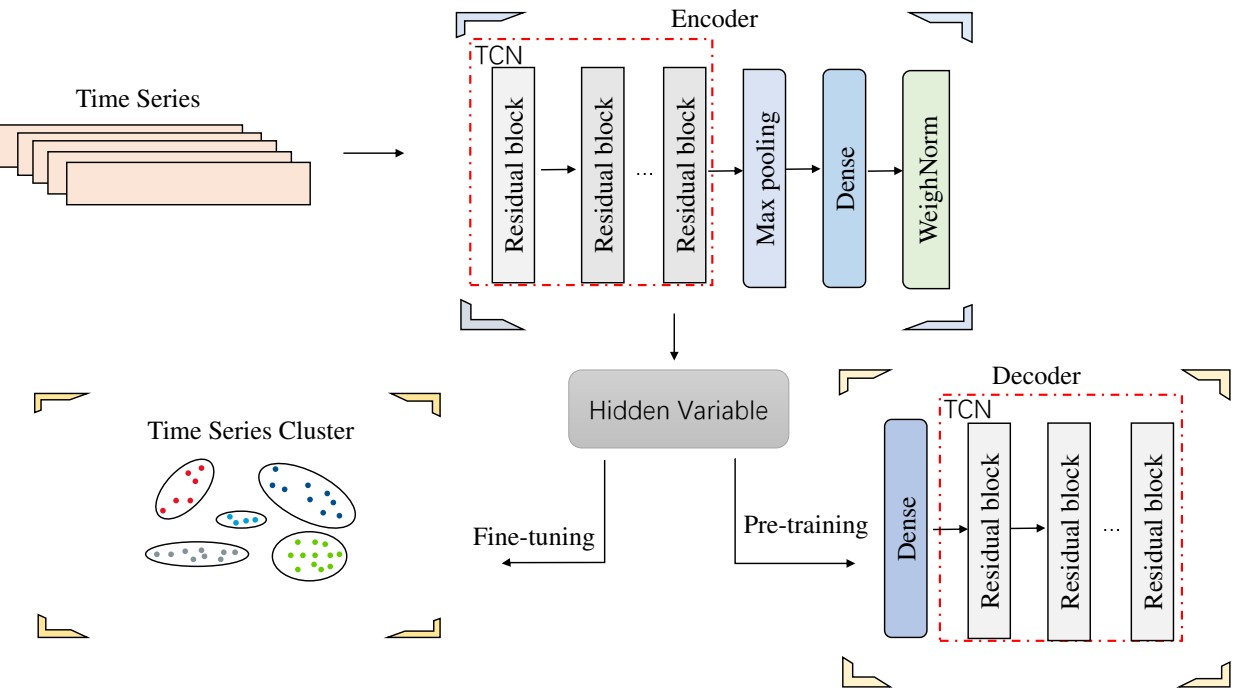

**Figure 1.** Architecture of TCN-Deep Clustering.

### 2.1. Autoencoder Network of TCN

To mine the features of financial time series, this paper designs an autoencoder based on a TCN network. The TCN network adds two operations compared to a 1D CNN: casual and dilated convolutions and a connection at the network level with a residual network, as shown in Figure 2. Each residual block contains two layers of convolutional and nonlinear operations (with ReLU as the activation function), and weightnorm and dropout layers are also added to regularize the network in each layer to avoid gradient disappearance caused by the training process. Unlike the fully connected 1D CNN shown in Figure 3a, causal convolution takes into account that the time series should only be affected by the current state or the past state, as shown in Figure 3b. To solve the problem of the limited receptive field of causal convolutional networks, a TCN network puts forward a dilated convolution. For a one-dimensional input time series $x = [x_1, x_2, ., x_t]$, $f : \{0, 1, 2, \ldots k - 1\}$ represents a convolution kernel. The dilated convolution at $x_t$ is calculated with Equation (1):

$$F(x_t) = (x *_d f)(x_t) = \sum_{i=0}^{k-1} i \cdot x_{t-d \cdot i} \tag{1}$$

where $k$ represents the convolution kernel size, $d$ represents the dilation factor, and $t - d \cdot i$ represents the past direction. For example, when $d = 1$, $k = 2$, $x_t$ performs the convolution operation with $x_{t-1}$ only. Figure 4 presents the TCN network structure with a convolution kernel of $k = 2$ and a dilation factor of $d = [1, 2, 4, 8]$. As shown in Figure 4, the third layer of the rightmost element can receive eight input elements. Without a dilated convolution, receiving an input of the same length causes the number of layers in the network to rise to eight, which leads to a dramatic increase in the number of parameters to be trained.

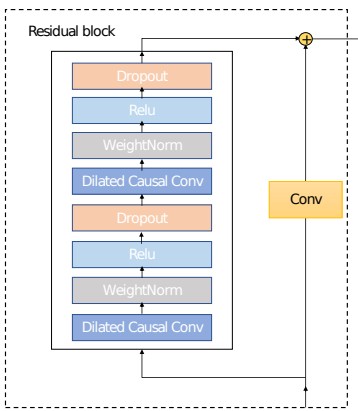

**Figure 2.** Residual Block.

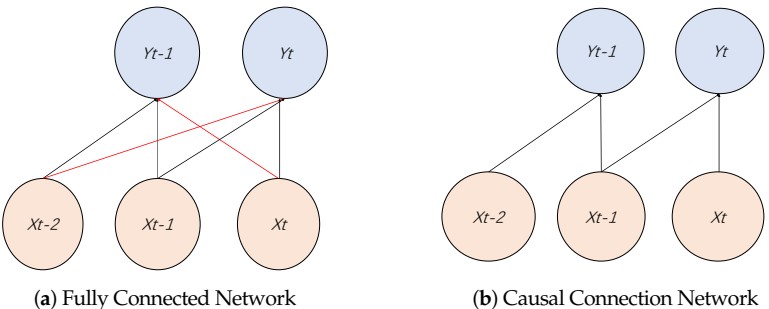

(**a**) Fully Connected Network          (**b**) Causal Connection Network

**Figure 3.** Two different types of network connections.

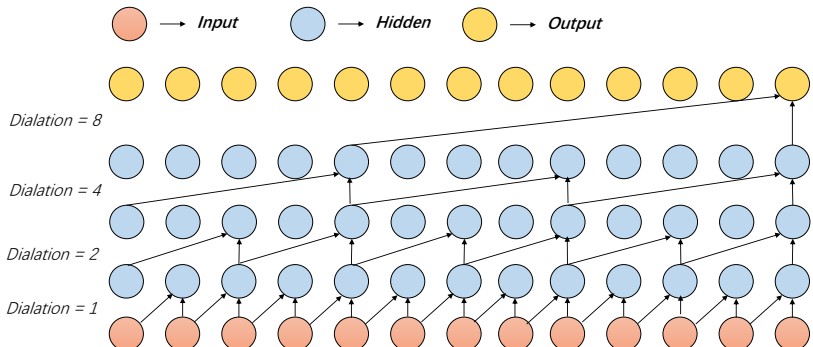

**Figure 4.** Casual Dilated Convolutional Network.

### 2.2. Training Process

The total loss of TCN-Deep Clustering consists of the sum of the losses of two components: the reconstruction loss and the KL divergence loss, as shown in Equation (10), which are presented next in terms of two temporal feature extraction and time-series clustering optimizations.

$$L_{TCN-\text{DeepClustering}} = L_{\text{r}} + L_{\text{c}} \tag{2}$$

#### 2.2.1. Time-Series Feature Extraction

Suppose $x = [x_1, x_2, \dots, x_t]$ is the original input stock time series. First, carry out the encoding process $f$, as shown in Figure 1. The purpose is the characteristic representation of the original data, the expressions of which are Equations (3) and (4), where $\sigma_f$ represents the

activation function operation in the encoding process, and the activation function used here is ReLU. $w$ and $b_1$ represent the convolution kernel weights and biases of the encoder process.

$$z_i = f(x) \tag{3}$$

$$z_i = \sigma_f(w * x + b_1) \tag{4}$$

The decoding process $g$, as shown in the right part of Figure 1, aims to reconstruct the features as the most similar representation of the original data, where $\sigma_g$ is the activation function of the deconvolution process and $w'$ and $b_2$ are the convolution kernel weights and biases in the decoding process, respectively.

$$x' = g(z_i) \tag{5}$$

$$x' = \sigma_g(w' * z_i + b_2) \tag{6}$$

The autoencoder is an unsupervised learning algorithm in which the loss function continuously adjusts the autoencoder parameters during the training process. It minimizes $L_r$ to make the reconstructed $x'$ close to $x$, which extracts the temporal feature $z_i$. The reconstruction loss is shown in Equation (7):

$$L_r = \frac{1}{n} \sum_{i=1}^{n} \|x - x'\|^2 \tag{7}$$

### 2.2.2. Time-Series Clustering Optimization

After completing the initial feature extraction phase of the TCN autoencoder, the decoder network of the autoencoder is lifted and only the encoder part of the feature extraction is retained, and the clustering layer is customized on this basis. Take the temporal feature $z_i$ extracted by the encoder as the input and obtain the stock clustering results through the similarity measurement (Euclidean distance). According to the optimal number of k clusters defined in the initial stage, the k cluster centers are obtained. The $i$-th cluster class center is $\mu_i$.

Traditional k-means uses the hard labeling approach to measure the similarity of the sample points to the cluster centers but it cannot measure the uncertainty of data to the cluster centers, especially if the outlier points are assigned with low accuracy, which may reduce the quality of clustering. Therefore, in this paper, the t-distribution of soft labels is used to measure the similarity between the sample points and clustering centers, as shown in Equation (8):

$$q_{ij} = \frac{\left(1 + |z_j - \mu_i|^2 / \alpha\right)^{-\frac{\alpha+1}{2}}}{\sum_i \left(1 + |z_j - \mu_i|^2 / \alpha\right)^{-\frac{\alpha+1}{2}}} \tag{8}$$

where $|z_j - \mu_i|^2$ is the distance from $z_j$ to the cluster center $\mu_i$. $q_{ij}$ is the is the probability of classifying $q_{ij}$ to the clustering center $\mu_i$. $\alpha$ is the degree of freedom of the t-distribution, which is set to 1. To improve the quality of clustering, the model defines a high-confidence target probability distribution based on the clustering center, as shown in Equation (9). $p_{ij}$ is the probability that $z_j$ is assigned to different clustering centers. $f_i$ represents the probability that all temporal features are assigned to clustering center $\mu_i$. $f_{i'}$ represents the probability that all temporal features are assigned to different clustering centers.

$$p_{ij} = \frac{q_{ij}^2 / f_i}{\sum_{i'} q_{i'j}^2 / f_{i'}}, f_i = \sum_j q_{ij} \tag{9}$$

In order to make the probability distribution of the soft distribution of the clustering layer $q_{ij}$ consistent with the auxiliary target distribution $p_{ij}$, the KL divergence is defined as the temporal clustering layer loss, as shown in Equation (10):

$$L_c = KL(P\|Q) = \sum_i \sum_j p_{ij} \log \frac{p_{ij}}{q_{ij}} \tag{10}$$

TCN-Deep Clustering trains the temporal clustering layer using the KL divergence, thereby optimizing the clustering centers $u_j$ and the encoder parameters of the autoencoder $\theta(w,b)$ simultaneously. The gradients of the loss function $L_c$ with respect to the temporal feature $z_i$ and the clustering center $u_j$ are calculated as (11) and (12).

$$\frac{\partial L_c}{\partial z_i} = \frac{\alpha+1}{\alpha} \sum_j \left(1 + \frac{|z_i - \mu_j|^2}{\alpha}\right)^{-1} (p_{ij} - q_{ij})(z_i - \mu_j) \tag{11}$$

$$\frac{\partial L_c}{\partial \mu_i} = -\frac{\alpha+1}{\alpha} \sum_i \left(1 + \frac{|z_i - \mu_j|^2}{\alpha}\right)^{-1} (p_{ij} - q_{ij})(z_i - \mu_j) \tag{12}$$

The gradient $\frac{\partial L_c}{\partial z_i}$ is transferred to the encoder network parameters and used for backpropagation to calculate the network parameter gradient $\frac{\partial L_c}{\partial \theta}$. In this study, the Adam optimization method was used to optimize the loss function.

### 2.3. TCN-Deep Clustering Algorithm

The TCN-Deep Clustering algorithm is shown in Algorithm 1. The model inputs include data set $D$, which includes a large amount of historical stock data; the number of clusters $K$; the maximum iteration $N$; and the pre-trained iteration $N_p$. The first row is initialized with k-means, which leads to the initial clustering center. The second and third rows pre-train the autoencoder network according to the reconstruction loss to obtain the initial network parameters $\theta$ and the temporal feature $z_j$. The fourth to ninth rows are trained iteratively according to the Adam [23], where the fifth row represents the encoder extracting the temporal features. The fifth and sixth rows calculate the distance between the features extracted by the encoder and the clustering centers and assign classes. The 9th and 10th rows calculate the target distribution and KL divergence. The 11th row indicates that the iterative process stops when the cluster assignment is less than $\delta$ for T consecutive iterative processes.

---

**Algorithm 1:** TCN-Deep Clustering Algorithm.

---

Model: TCN-Deep Clustering

Input: Data set $D$; Number of clusters $K$; Maximum iteration $N$; Pre-trained
  iteration $N_p$;

Output: Clustering results $S$;

1 K-means initialize the clustering center $\mu = (\mu_1, \mu_2 \ldots \mu_k)$

2 for $i = 1$ to $N_p$:

3      $L_r = \frac{1}{n} \sum_{i=1}^{n} \|x - x'\|^2$, autoencoder preliminary feature extraction.

4 for $i = 1$ to $N$:

5      Encoder extraction of input features: $z_j = f(x_j)$

6      Calculating the soft distribution $p_{ij}$ of $z_j$ with cluster class center $\mu_i$

7      Assign input data to clusters; Obtain $S$

8      Calculating the target distribution $q_{ij}$

9      Calculating $L_c = KL(P\|Q) = \sum_i \sum_j p_{ij} \log \frac{p_{ij}}{q_{ij}}$

10     Based on Adam, to update cluster centers $u_i$ and encoder parameters $\theta$

11 end if ($i\ Mod\ T == 0$)&& Cluster assignment is less than the threshold $\delta$

---

## 3. Experimental Settings and Results

### 3.1. Data Description

This paper focuses on the closing prices of two important index markets in China and the United States, the CSI 300 and the S&P 500 index. The historical ticker data of the CSI 300 index comes from the Joinquant platform and the historical ticker data of the S&P 500 index comes from Yahoo Finance. The collected data are shown in Table 1. Stocks with long-term suspensions and ST stocks were removed from the paper. The time period for the data collected in the experiment was from 1 January 2015 to 1 July 2018. We divided the time period into three parts. The first part was the training set; this period was mainly used to define the optimal number of clustering classes and the selection of the hyperparameters, as well as the application to obtain the clustering results. The second time period was the observation set, where the market return analysis was conducted for a similar set of different stocks that was obtained in the previous step. The third time period was the backtesting phase, where the empirical analysis was conducted mainly by the stock selection scheme obtained from the analysis. Figure 5a,b show the closing price movements of randomly selected constituents from the S&P 500 and CSI 300 indices, respectively.

**Table 1.** Statistics of research data.

| Index | Stocks | Training Set 01/01/2015 01/03/2018 | Observation Set 01/03/2018 01/06/2018 | Simulation Set 01/06/2018 01/07/2018 |
|---|---|---|---|---|
| CSI 300 | 290 | 1155 | 92 | 30 |
| S&P 500 | 478 | 1155 | 92 | 30 |

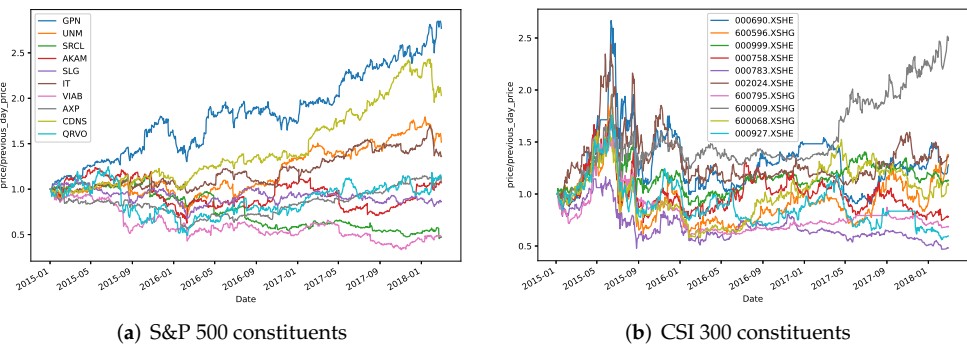

(**a**) S&P 500 constituents          (**b**) CSI 300 constituents

**Figure 5.** The closing price movements of randomly selected constituents from the S&P 500 (**a**) and CSI 300 (**b**) indices.

### 3.2. The Number of Clusters

This experiment used k-means to initialize the model clustering groups. As the number of clusters K increased, the samples were divided further and the degree of aggregation of each cluster gradually increased, and then the SSE naturally decreased gradually. Below, $C_i$ represents the $i$th cluster, $p$ represents the sample points of the corresponding cluster, $m$ represents the center of mass of the corresponding cluster, and SSE is the clustering error of all clustered samples, which is used to represent the clustering performance. The definition is given in the following equation:

$$SSE = \sum_{i=1}^{k} \sum_{p \in C_i} |p - m_i|^2 \qquad (13)$$

The core idea of the elbow method is the relationship between the corresponding $k$-value and its corresponding SSE, and the rate of decrease of SSE will be reduced when $k$ is close to the optimal number of clusters. Then, by increasing $k$ and SSE, the decline tends

to smooth out, as shown in Figure 6. Finally, $k = 150$ and 100 were selected as the optimal number of clusters for the S&P 500 and CSI 300 markets, accordingly.

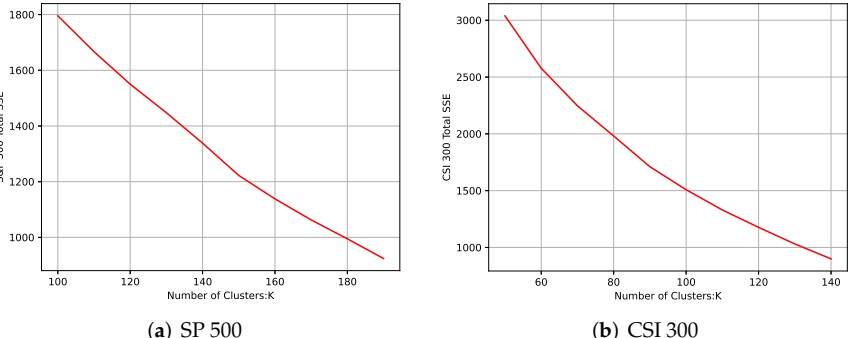

(**a**) SP 500         (**b**) CSI 300

**Figure 6.** The relationship between SSE and K.

### 3.3. Parameter Settings

The network parameters of the TCN encoder–decoder structure for this experiment are shown in Table 2. The features of the input layer used the stock closing price divided by the closing price of the previous day, which was then fed into the encoding and decoding layer structure of the network. The encoding layer corresponds to rows two to eight in Table 2, which mainly performed a dimension-raising operation on the data. Furthermore, the decoding layer corresponds to rows ten to eighteen in Table 2, which mainly performed a dimension-lowering operation on the data. The hidden state dimension was set to 320 dimensions. The number of iterations of the encoder was set to 100 and the Adam optimizer was adopted during the training process. Meanwhile, the initial learning rate was set to 0.001. The total number of parameters to be trained for the network reached 309,644.

**Table 2.** TCN encoder–decoder network structure and parameter settings.

| No | Type | Layer | Output Shape | Parameters |
|----|------|-------|--------------|------------|
| 1 | —— | Input_1 (InputLayer) | (None, None, 1) | 0 |
| 2 | Encoder (Layer 1) | residual_block_1 | (None, None, 40) | 5162 |
| 3 | Encoder (Layer 2) | residual_block_2 | (None, None, 40) | 9762 |
| 4 | Encoder (Layer 3) | residual_block_3 | (None, None, 40) | 9762 |
| 5 | Encoder (Layer 4) | residual_block_4 | (None, None, 40) | 9762 |
| 6 | Encoder (Layer 5) | residual_block_5 | (None, None, 40) | 9762 |
| 7 | Encoder (Layer 6) | residual_block_6 | (None, None, 160) | 103,202 |
| 8 | Encoder (Layer 7) | global_max_pooling1d | (None,160) | 0 |
| 9 | Encoder (Layer 8) | dense | (None, 320) | 51,520 |
| 10 | Encoder (Layer 9) | activation | (None, 320) | 0 |
| 11 | Decoder (Layer 10) | dense | (None, 796) | 255,516 |
| 12 | Decoder (Layer 11) | reshape | (None, 796, 1) | 0 |
| 13 | Decoder (Layer 12) | decoder_residual_block_1 | (None, 796, 40) | 5162 |
| 14 | Decoder (Layer 13) | decoder_residual_block_2 | (None, 796, 40) | 9762 |
| 15 | Decoder (Layer 14) | decoder_residual_block_3 | (None, 796, 40) | 9762 |
| 16 | Decoder (Layer 15) | decoder_residual_block_4 | (None, 796, 40) | 9762 |
| 17 | Decoder (Layer 16) | decoder_residual_block_5 | (None, 796, 40) | 9762 |
| 18 | Decoder (Layer 17) | decoder_residual_block_6 | (None, 796, 1) | 170 |

### 3.4. Evaluation Indicators

The evaluation methods for clustering results can generally be divided into internal and external evaluation methods. An external evaluation evaluates the goodness of the clustering results when the true labels are known, and since there were no true labels for the stock data, an internal evaluation was used. For a clustering method, a lower intra-cluster aggregation and higher inter-cluster coupling can indicate a better performance of the

clustering method. In this paper, the SC (Silhouette Score) [24], CH (Calinski–Harabaz) [25], and DB (Davies–Bouldin) [26] were adopted to evaluate the clustering performance.

### 3.4.1. SC

The SC is a way of evaluating the effectiveness of clustering. The SC takes a range of $[-1,1]$, and the larger the SC, the better the clustering performance. The definitions are defined in Equations (14) and (15).

$$\text{sc}_i = \frac{b_i - a_i}{\max(a_i, b_i)} \tag{14}$$

$$SC_{\text{total}} = \frac{1}{N} \sum_{i=1}^{N} sc_i \tag{15}$$

Equation (14) represents the SC for a single sample, where $a_i$ represents the average distance between the cluster to which the ith sample belongs and the other samples in the same cluster. If there is only one sample in the cluster, $sc_i = 0$. $i \in A, a_i = \text{average}_{j \in A, i \neq j}(\text{dist}(i,j))$, where $A$ represents one of the K clusters. $b_i$ represents the minimum value of the average distance between the ith sample and the other clusters. $i \in A, C \neq A, \text{dist}(i,C) = \text{average}_{j \in C}(\text{dist}(i,j))$. $b_i = \min_{C \neq A} \text{dist}(i,C)$.

### 3.4.2. CH

The CH is also an indicator for evaluating good or bad clustering effects that is calculated much faster than the SC, and is calculated as follows.

$$CH = \frac{SS_B}{K-1} / \frac{SS_W}{N-K} \tag{16}$$

$$SS_B = \text{tr}(B_k), B_k = \sum_{q=1}^{k} n_i (c_q - c_e)(c_q - c_e)^T \tag{17}$$

$$SS_W = \text{tr}(W_k), W_k = \sum_{q=1}^{k} \sum_{x \in C_q} n_i (x - c_q)(x - c_q)^T \tag{18}$$

where $SSB$ is the variance between clusters and $SSW$ is the variance within clusters. The ratio of compactness to separation is the $CH$ index, where $c_q$ represents the center point of cluster $q$, $c_e$ represents the center point of the data set, $n_q$ represents the number of data in cluster $q$, and $C_q$ represents the data set of cluster $q$. The sum of the square distances between each cluster point and its center is used to determine the cluster's compactness. The sum of the square distances between the cluster centers and the center of the data set as a whole is used to determine the degree of separation between the datasets. A higher $CH$ index indicates improved clustering efficiency.

### 3.4.3. DB

The DB combines intercluster distance and intracluster dispersion to determine the performance of the clustering algorithm, which is calculated as follows:

$$DB = \frac{1}{K} \sum_{k=1}^{K} \max_{k'=1,\dots,K, k' \neq k} \left( \frac{\sigma_k + \sigma_{k'}}{d_{kk'}} \right) \tag{19}$$

where $\sigma_k$ is the intracluster dispersion of the kth cluster, $d_{kk'} = |u_k - u_{k'}|$, and $d_{kk'}$ is the intercluster distance between the kth and $k'$ clusters, where $u_k$ and $u_{k'}$ represent the distance between the $k$ and $k'$ cluster class centers, respectively. A small $DB$ means a nice clustering performance.

### 3.5. Results and Analysis

In order to verify the effectiveness of the proposed method, a series of experimental metrics of the clustering aggregation degree was obtained by comparing the proposed method with general clustering methods as well as deep clustering methods. Two categories of clustering algorithms were used for comparison to verify the effectiveness of TCN-Deep Clustering. The first category included the general k-means, k-shape, spectral, etc. These algorithms only utilize the original features of the data and the temporal dependences of the data are not mined deeply. To verify that the clustering performance can be improved by deep neural networks, several comparative experiments were implemented. The second category adopted a deep clustering algorithm, of which DEC is a typical algorithm. It can also optimize clustering performance by feature extraction but is insensitive to financial time series.

The proposed method was verified in two index markets and the performance is shown in detail in Tables 3 and 4. In general, deep clustering algorithms outperformed general algorithms in terms of clustering aggregation. Although k-means performed best among the general clustering algorithms, its clustering aggregation was still much smaller than that of deep clustering algorithms. Meanwhile, spectral performed worst out of all of the clustering algorithms. As for feature extraction, DEC performed worse than TCN-Deep Clustering in terms of the CH and DB in the S&P 500 index, whereas the SC had only a slight superiority. In contrast, TCN-Deep Clustering performed best in the case of the evaluation metrics in the CSI 300 index.

**Table 3.** S&P 500 Index clustering results.

| Indicator | General Clustering | | | Deep Clustering | |
|:---:|:---:|:---:|:---:|:---:|:---:|
| | **K-Means** | **K-Shape** | **Spectral** | **DEC** | **TCN-Deep Clustering** |
| SC | 0.1088 | 0.1086 | 0.1021 | **0.1256** | 0.1168 |
| CH | 116.8139 | 114.3262 | 107.0936 | 120.6435 | **130.9683** |
| DB | 1.0986 | 1.1563 | 1.1595 | 1.0553 | **0.9360** |

**Table 4.** CSI 300 Index clustering results.

| Indicator | General Clustering | | | Deep Clustering | |
|:---:|:---:|:---:|:---:|:---:|:---:|
| | **K-Means** | **K-Shape** | **Spectral** | **DEC** | **TCN-Deep Clustering** |
| SC | 0.1272 | 0.1215 | 0.1023 | 0.1315 | **0.1329** |
| CH | 54.0336 | 54.5807 | 34.3582 | 51.1134 | **63.68** |
| DB | 0.9992 | 1.0509 | 1.0295 | 0.9162 | **0.8175** |

To further illustrate the clustering effectiveness of TCN-Deep Clustering, the following visualization is shown to explain the rationality behind the clustering shown in Figures 7 and 8. Four different clusters were randomly selected from each of the two markets. As shown in these figures, TCN-Deep Clustering clustered stocks with similar price trends. Taking the S&P500 index as an example, TCN-Deep Clustering was able to categorize them well during trending-up, trending-down, and oscillating markets. The cluster in Figure 7c, for example, contains two companies, GOOG and GOOGL, which operate similar businesses and have the same price trends; however, this cluster also includes non-technology stocks such as EW and FLSV, which also shows in one way that stock prices are affected by uncertainties among different industries.

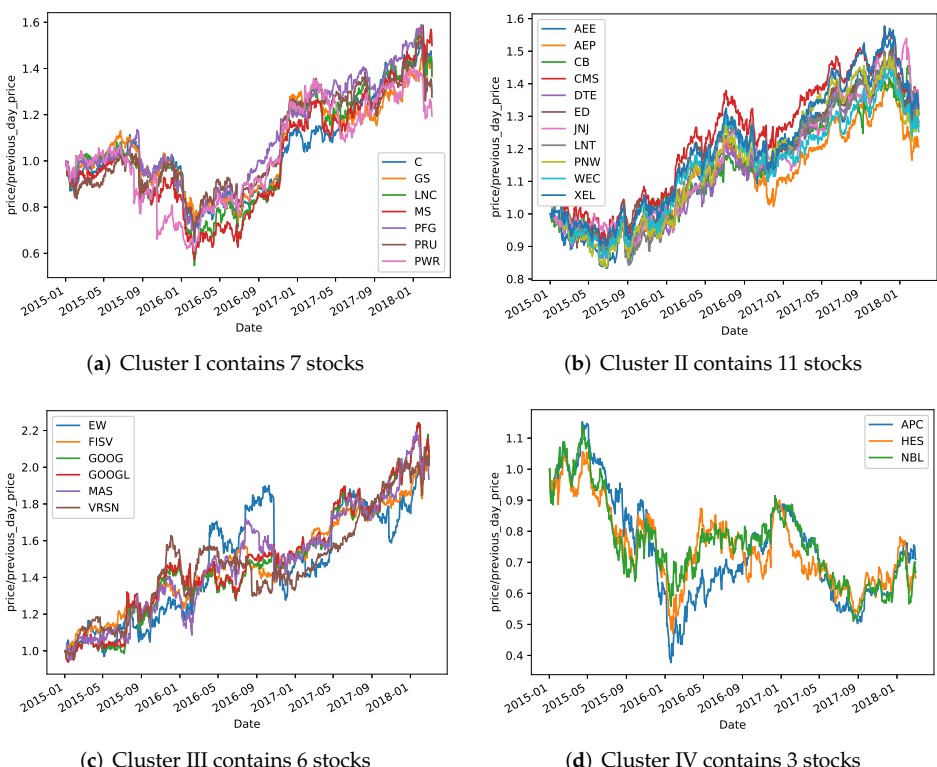

(**a**) Cluster I contains 7 stocks

(**b**) Cluster II contains 11 stocks

(**c**) Cluster III contains 6 stocks

(**d**) Cluster IV contains 3 stocks

**Figure 7.** Randomly selected groups from S&P500 based on TCN-Deep Clustering.

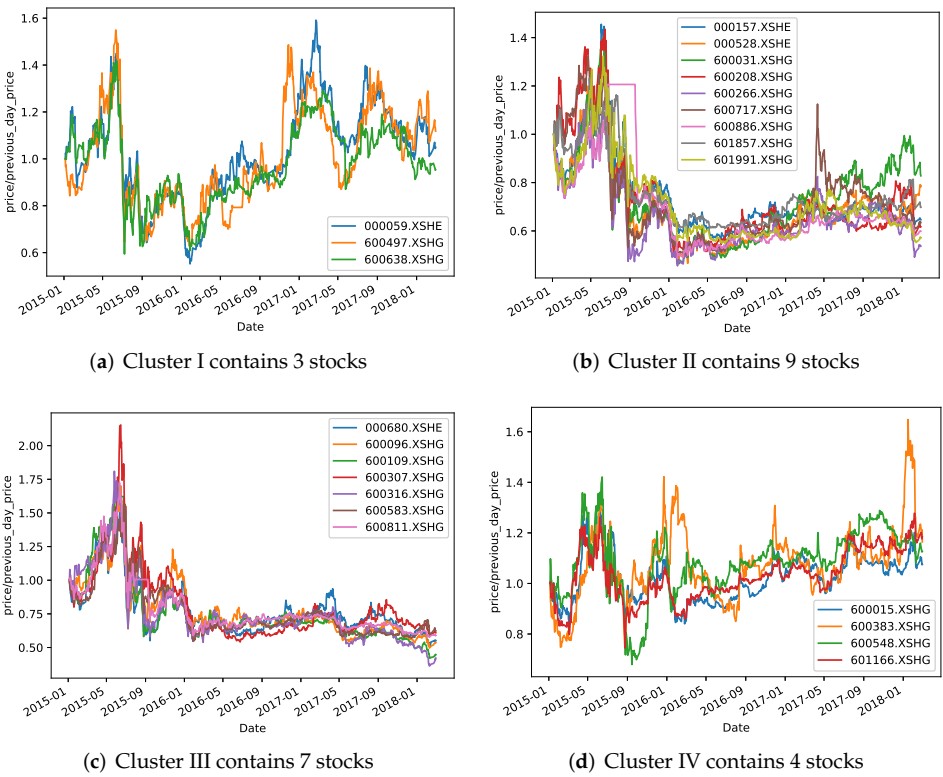

(**a**) Cluster I contains 3 stocks

(**b**) Cluster II contains 9 stocks

(**c**) Cluster III contains 7 stocks

(**d**) Cluster IV contains 4 stocks

**Figure 8.** Randomly selected groups from CSI 300 based on TCN-Deep Clustering.

## 4. Backtest Analysis

In this section, the clustering results of multiple stocks are analyzed using the clustering algorithm proposed. Based on the combination of the clustering results of TCN-Deep

Clustering and the historical performance of individual stocks, a number of representative specific stocks are screened. Thus, a clustering-based stock selection strategy is constructed, which illustrates the guiding significance of the clustering algorithm proposed in this paper in the financial context.

*Construction of Stock Selection Strategy Based on Clustering Results*

The stock selection strategy revolved around a hypothesis: stocks within the same cluster have similar price movements and achieve similar returns over a given period of time. During the observation period, we first selected stocks that were above or below average by setting different thresholds. During the testing period, we bought stocks based on those above average and sold stocks based on those below average, as shown in Figure 9.

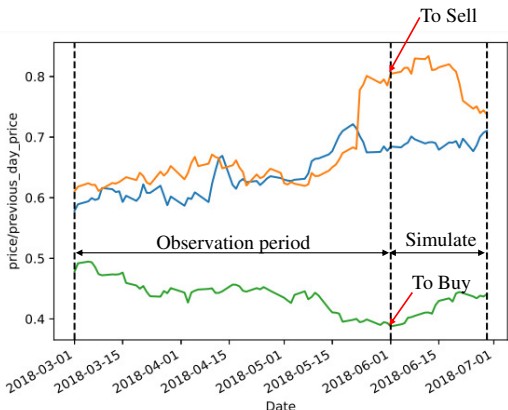

**Figure 9.** Visual explanation of buying and selling.

By selecting the stocks we needed to buy and sell in the testing period, we learned that in most cases, the stocks selected had higher returns than the market average returns, for example, in the S&P 500 index market shown in Table 5, in the cases where the thresholds were set to 0.02, 0.04, 0.06, 0.08, 0.1, the stocks to buy revealed higher returns than the market average returns. The strategy outperformed the market average by 0.77% when the threshold was set to 0.1. Similarly, in the CSI 300 index market shown in Table 6, the return reached a maximum of 2.41% (annual return can reach 28.89%) when the threshold was set to 0.02. Of course, there is some uncertainty in the market, for example, in the S&P 500 index market when the threshold was set to 0.02, the return to buy was lower than the average market return, and this was also reflected in the CSI market.

**Table 5.** Returns in S&P 500.

| S&P 500 | Different Thresholds | | | | |
|---|---|---|---|---|---|
| | Th = 0.02 | 0.04 | 0.06 | 0.08 | 0.1 |
| To buy | 1.0113 | 1.0068 | 1.0081 | 1.0077 | 1.0125 |
| To sell | 1.0070 | 1.0025 | 1.0066 | 0.9899 | 0.9866 |
| Average | 1.0048 | 1.0048 | 1.0048 | 1.0048 | 1.0048 |

**Table 6.** Returns in CSI 300.

| CSI 300 | Different Thresholds | | | | |
|---|---|---|---|---|---|
| | Th = 0.02 | 0.04 | 0.06 | 0.08 | 0.1 |
| To buy | 0.9455 | 0.9192 | 0.9253 | 0.9290 | 0.9476 |
| To sell | 0.9135 | 0.9454 | 0.9256 | 0.9104 | 0.9245 |
| Average | 0.9214 | 0.9214 | 0.9214 | 0.9214 | 0.9214 |

From the above results, it can be verified that TCN-Deep Clustering can be combined with a stock selection strategy based on historical price performance to obtain stocks that

outperform the average returns. Specifically, investors can select a period in the past for clustering analysis and choose the corresponding threshold to invest according to their investment risk tolerance, at which time the corresponding buying pool and selling pool will appear, corresponding to stocks in the buying pool that can be bought or added to and stocks in the selling pool that can be sold or reduced.

## 5. Conclusions

In this paper, we propose a new deep temporal clustering algorithm defined as TCN-Deep Clustering, which exploits the important feature extraction capability of TCN networks in the field of time series combined with the clustering loss of deep clustering network algorithms to solve the difficulties in the manual feature extraction of general clustering algorithms and the insensitivity of existing deep clustering networks to time-series features. The experimental objects are the important index components of two countries, China and the USA. It is verified by the experimental results that the proposed method can find stocks with similar price trends and outperforms the general clustering algorithm and deep clustering algorithm to some extent. To further validate the practicability of the study, an inference stock selection strategy is constructed by selecting stocks that perform well or poorly in the same cluster. Through this stock selection strategy, the proposed method can achieve a good return in the actual market. By excelling in the field of finance, the proposed method could also achieve significant results in other time-series fields.

**Author Contributions:** Data curation, M.L.; Methodology, M.L.; Resources, Y.C.; Writing—original draft, Y.C. and M.L.; Writing—review and editing, M.L., G.C., C.Z. and Z.C. All authors have read and agreed to the published version of the manuscript.

**Funding:** This research was funded by the general scientific research projects of the Zhejiang Provincial Department of Education (Y202249481).

**Data Availability Statement:** The data that support the findings of this study are available from the corresponding author, Y.C., upon reasonable request

**Acknowledgments:** We are grateful to the Joinquant and Yahoo Finance platform for their dataset.

**Conflicts of Interest:** The authors declare no conflicts of interest. The funders had no role in the design of the study; in the collection, analyses, or interpretation of data; in the writing of the manuscript; or in the decision to publish the results.

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
