# Peer review of "Financial Market Correlation Analysis and Stock Selection Application Based on TCN-Deep Clustering"

_futureinternet, doi:10.3390/fi14110331_

Round 1

Reviewer 1 Report

The article is very interesting. Concerns the current problem.
In this paper, Authors propose a new deep temporal clustering algorithm defined as TCN- DeepClustering which exploits the important feature extraction capability of TCN in the field of time series, combined with the clustering loss of deep clustering network algorithms, to solve the difficulties of manual feature extraction of general clustering algorithms and the insensitivity of existing deep clustering networks to time series features.
The innovative method proposed by the authors includes two modules:  an autoencoder feature extraction network based on TCN (Temporal Convolutional Networks) and a temporal clustering optimization algorithm with KL(Kullback-Leibler) divergence. The features of financial time series are represented by causal convolution and dilated convolution of TCN networks. Then the pre-training results based on KL divergence are fine-tuned to make the clustering results discriminative.
The article is written carefully and clearly.

Reviewer 2 Report

The topic per se is very interesting and the authors have done a reasonable job of conducting the empirical work. However, it has potential to become more interesting.

I have only the following comments.

a)      The presentation and structure of this paper is correct and enjoyable to read. The idea of the paper is quite innovative.

b)      Moreover, the econometric methodology is appropriate; an other advanced econometric technique that analyzes the co movements / spillover effects is the continuous wavelet transformation (CWT). The author should examine the possibility of adopting this method (as a robustness test) or at least make a reference to it. Please see the following paper:

 -          Dimitrios Dimitriou, Dimitris Kenourgios, Theodore Simos (2020). “Are there any other safe haven assets? Evidence for “exotic” and alternative assets”. International Review of Economics & Finance, Vol. 69, 614-628.

c)      Finally, the key results should be interpreted in a way that will provide guidance to fund managers and investors.

Reviewer 3 Report

The article requires a major linguistic correction. There are typos that make the paper literally illegible. For example, I can understand that "elbow method" (line 146) is a typo ("method below"). But I cannot understand what the Authors meant by "the shape of a hand elbow" (line 148).

The methods presented in the paper seem interesting, but the presentation needs serious improvement: linguistic and mathematical. Based on the presented description I completely cannot say, how the presented model works. Many symbols in the equations do not have explanations. For example, what is $p_{ij}$ and $q_{ij}$ in equation (4)? Latter (line 109) it is explained that "p denotes the target distribution", but it is not explained what is this "target distribution". I have also doubts concering the equations (5) and (6). There should be the derivatives of the loss function $L_c$, but using standard calculus rules I do not obtain this. It can be a problem connected with the lack of explanations of the symbols in eqn. (4).

To sum it up: the article requires a major revision and the main problem is the poor presentation of the method.

Reviewer 4 Report

A file with the report is attached.

Round 2

Reviewer 2 Report

The comments addressed properly.

Reviewer 3 Report

The article was corrected, however it still requires linguistic and editing corrections. Especially there are many mistakes with the punctation (egg. lack of space after dots or commas).

Reviewer 4 Report

A file with the report is attached.
